# Comparison of Palliative Care Models in Idiopathic Pulmonary Fibrosis

Sarah Younus [1], Jeffrey A. Bakal [2], Janice Richman-Eisenstat [1,3], Ghadah Alrehaili [4], Sharina Aldhaheri [4], Michelle Morales [5], Naomi Rippon [5], Elisabeth Bendstrup [6], Ingrid Harle [7], Onofre Moran-Mendoza [4], Shaney L. Barratt [5], Huzaifa Adamali [5] and Meena Kalluri [1,3,*]

1   Division of Pulmonary Medicine, Department of Medicine, University of Alberta, Edmonton, AB T6G 2G3, Canada; syounus@ualberta.ca (S.Y.); janice.richmaneisenstat@albertahealthservices.ca (J.R.-E.)
2   Provincial Research Data Services, Alberta Health Services, Edmonton, AB T6G 2C8, Canada; jbakal@ualberta.ca
3   Alberta Health Services, Edmonton, AB T6G 1Z1, Canada
4   Division of Respirology and Sleep Medicine, Department of Medicine, Queen's University, Kingston, ON K7L 2V6, Canada; 17gaa2@queensu.ca (G.A.); aldhahes@kgh.kari.net (S.A.); morano@queensu.ca (O.M.-M.)
5   Bristol Interstitial Lung Disease, North Bristol Trust, Southmead Hospital, Bristol BS10 5NB, UK; Michelle.Westlake@nbt.nhs.uk (M.M.); Naomi.Rippon@nbt.nhs.uk (N.R.); Shaney.Barratt@nbt.nhs.uk (S.L.B.); Huzaifa.Adamali@nbt.nhs.uk (H.A.)
6   Center for Rare Lung Diseases, Department of Respiratory Diseases and Allergy, Aarhus University Hospital, 8200 Aarhus, Denmark; karbends@rm.dk
7   Division of Palliative Care, Department of Medicine, Queen's University, Kingston, ON K7L 3J7, Canada; iaharle@xplornet.ca
*   Correspondence: kalluri@ualberta.ca

**Abstract:** Introduction: Palliative care (PC) is recommended in idiopathic pulmonary fibrosis (IPF) patients but poorly implemented. Integration of PC into routine management by pulmonologists may improve overall and end-of-life (EOL) care, but the optimal model of PC delivery is unknown. Objective: To describe three PC care delivery models and their impact on EOL; the Multidisciplinary Collaborative ILD clinic, Edmonton, Canada (EC) and the Bristol ILD Service, UK (BC) that provide primary level PC; and the Queen's University ILD Clinic, Kingston, Canada (QC), which refers IPF patients to a specialist PC Clinic using specific referral criteria. Methods: A multicenter retrospective observational study of IPF patients receiving care in the identified clinics (2012–2018) was designed. Demographics; PC delivery, including symptom management; advance care planning (ACP); and location of death data were examined. Results: 298 IPF patients were included (EC 95, BC 84, and QC 119). Median age was 71 years with 74% males. Overall, 63% (188) patients received PC. Primary PC approach in EC and BC led to more patients receiving PC (98% EC, 94% BC and 13% QC ($p < 0.001/<0.001$)) with earlier initiation compared to QC. Associated higher rates of non-pharmacologic dyspnea management [98% EC, 94% BC, and 2% QC ($p < 0.001/<0.001$); opioids (45% EC and BC, and 23% QC ($p < 0.001/<0.001$)); and ACP (100% EC and BC, and 13% QC patients ($p < 0.001/<0.001$))] were observed. Median follow up (IQR) was 16 months (5–28) with 122 deaths (41%). Primary PC model in EC and BC decedents was associated with more PC delivery (91% EC, 92% BC and 19% QC ($p < 0.001$)) with more symptoms management, oxygen, and opiate use than QC ($p < 0.001$; $p = 0.04$; $p = 0.01$). EOL discussions occurred in 73% EC, 63% BC, and 4% QC decedents ($p = 0.001$). Fifty-nine% (57) died at home or hospice and 38% (36) in hospitals. Concordance rate between preferred and actual location of death was 58% in EC (0.29 (−0.02–0.51)) and 37% in BC models (−0.11 (−0.20–0.15)). Conclusions: Primary PC approach for IPF is feasible in ILD clinics with concurrent disease management and can improve access to symptom management, ACP, PC and EOL care. Reliance on PC specialist referral for PC initiation outside of the ILD clinic can result in delayed care.

**Keywords:** palliative care; integrated; ambulatory; care models; IPF; advance care planning; end-of-life care

## 1. Introduction

Patients with idiopathic pulmonary fibrosis (IPF) and their caregivers experience a high symptom burden, inadequate preparation for end-of-life (EOL), and lack of access to community supports, resulting in patient suffering due to poor care and health-related quality of life (HRQol) [1,2]. The aim of palliative care (PC) is to relieve suffering and improve HRQol for patients and their families affected by advanced illness. Therefore, clinical guidelines and expert consensus recommendations are to integrate PC early in the course of IPF given high needs, and in impaired HRQol up to 2 years before death [3–5]. Unfortunately, the palliative care needs of this patient population often remain unmet due to a lack of knowledge regarding symptom management and the optimal models of service delivery, and a culture of neglect among practitioners [5] despite calls for change [6,7]. While the impact of PC on HRQol in IPF is unknown at this time, preliminary evidence suggests that early integration of PC into routine management may improve symptom management, HRQol, and end-of-life (EOL) care by meeting patient preferences and reducing unscheduled healthcare use and associated costs [8,9]. The optimal model of service delivery and standards for PC interventions have yet to be determined for IPF patients [10]. Traditionally, patients with needs are identified and referred to PC experts; this care model is called specialist PC. However, given the increase in demand and shortage of PC experts, it is strongly recommended that all practitioners develop competency in PC to address such unmet needs of the patient populations [5,11]. This model of PC delivered by professionals without dedicated training in PC is generalist or primary palliative care. The latter model is increasingly preferred given the growing population of chronic diseases with PC needs [12]. There are limited descriptions of primary PC models in ILD.

We describe three clinic models of PC delivery for IPF patients two located in Canada and one in the UK. These three models represent a spectrum of PC delivery ranging from 100% primary to 100% specialist PC, including a hybrid model that involves PC multidisciplinary discussion where PC specialist input is sought when needed [13]. Specifically, we aimed to describe and compare the components of PC delivered within each model, the time to initiation of symptom therapies, engagement in advanced care planning (ACP) discussions, and the impact of PC delivery on EOL care.

## 2. Methods

This is a multicenter, international, retrospective observational study of IPF patients seen at 3 sites: The Multidisciplinary Collaborative ILD Clinic in Edmonton, Canada (EC) and the Bristol ILD Service Clinic, UK (BC), which provide primary level PC; and the Queen's University ILD Clinic, Kingston, Canada (QC), which refers IPF patients to a specialist PC Clinic using specific referral criteria. Further descriptions of the clinics are provided in Supplement Table S1. The three clinics involved in this study developed their strategies independently. The medical records of consecutive patients with a diagnosis of IPF (2011 ATS) who attended clinics at the three sites were reviewed: EC (2012–2018), BC (2016–2018), and QC (2014–2018). This period corresponds to the start of the respective PC model at each site. IPF patients with missing data (patient moved/data not recorded) and non-IPF patients were excluded from the study.

Baseline patient characteristics were recorded, including demographics, symptoms, medications, lung function tests, and 6-min walk distance (6MWD). We extracted the following data from the first ILD clinic visit until censoring date on 30 June 2018, discharge from clinic, or death: (1) non-pharmacological and pharmacological symptom therapies, (2) ACP documentation and caregiver engagement, and (3) location and cause of death if known for decedents. Patients were considered to have received PC if they were provided symptom management (non-pharmacologic, oxygen, or opiates) and ACP. Details are provided in Supplement Table S1. The Health Research Ethics Boards in Alberta (REB Pro00082981), Ontario (HSREB 6025437), and Bristol, UK (IRAS 254044) approved the study.

*Statistical Analysis*

The data were summarized as medians and interquartile ranges (IQR) and percentages as appropriate. Statistical comparisons were made using Kruskal–Wallis rank tests and Fisher exact or chi-square tests as appropriate. Agreement on location of death versus patient-stated preference was summarized as ratios, and Cohen's Kappa was used to assess the agreement between the preferred and recorded location of death. All tests with a $p$-value $\leq 0.05$ were considered statistically significant. All analyses were done using Excel and R4.0 (Vienna, Austria).

## 3. Results

### 3.1. Patient Characteristics

Baseline demographic and clinical characteristics are presented in Table 1; 298 consecutive IPF patients were included (EC $n = 95$, BC $n = 84$, QC $n = 119$). The median (IQR) age was 71 years (67–78), and 74% (220) were males with BMI 28.3 kg/m$^2$ (26–32) and Charlson comorbidity index of 4 (3–5). Dyspnea and cough were reported in 97% and 82%, respectively, with median MRC 3 (2–4). The median FVC was 73% (61–85%), and DLCO was 48% (39–61%), with DLCO < 35% assessed in 17% (40) of patients. The median 6MWD was 330 m (240–424). GAP Stage II or higher was reported in 63% (175) of the cohort. More patients in EC (22%) and BC (31%) were in GAP stage III compared to QC (9%). Antifibrotic therapies were used in 40% of patients at baseline and 63% (187) by end of study (Supplement Table S2). Median follow up time in ILD clinic was 16 (5–28) months (Supplement Table S3). One patient was transplanted, and nine were on the wait list, suggesting use of concurrent life-extending therapies with PC.

**Table 1.** Demographics.

| Patient Profile at Baseline | All Cohorts (*n* = 298) | EC (*n* = 95) | BC (*n* = 84) | QC (*n* = 119) |
|---|---|---|---|---|
| Age, years (median, IQR) | Median: 71 (IQR 67–78) | Median: 71 (IQR 65–76) | Median: 74 (IQR: 68–78) | Median: 74 (IQR 68–79) |
| Gender (%) | M: 73.8% (220), F: 26.1% (78) | 65.3% m (62), 34.7% f (33) | 85.7% m (72), 14.3% f (12) | 72.3% M (86), 27.7% F (33) |
| BMI (median, IQR) | 28.3, 25.5–32.0 | 29.0, 26.0–33.3 | 26.4, 24.6–28.7 | 29.9, 26.5–32.2 |
| CCI (median, IQR) | 4, 3–5 | 3, 3–4 | 4, 3–5 | 4, 3–5 |
| Ever smoker (%) | 78.1% (225/288) | 82.0% (73/89) | 77.1% (64/83) | 75.9% (88/116) |
| Dyspnea (Y/N %) | 97.0% (289/298) | 95.8% (91/95) | 96.4% (81/84) | 98.3% (117/119) |
| MRC score (median, IQR) | 3, 2–4 | 3, 2–4 | 3, 2–4 | 2, 2–4 |
| Cough (Y/N %) | 81.6% (239/293) | 79.8% (75/94) | 75.9% (63/83) | 87.1% (101/116) |
| Fatigue (Y/N %) | 32.1% (86/268) | 52.9% (37/70) | 27.2% (22/81) | 23.1% (27/117) |
| % pred FVC (median, IQR) | 73%, 61–85% | 73%, 63–83% | 69%, 57–80% | 81%, 70–92% |
| FVC $\leq$ 50% | 10.1% (28/278) | 9.20% (8/87) | 13.1% (11/84) | 8.41% (9/107) |
| % pred DLCO adj for Hg FVC (median, IQR) | 48%, 39–61% | 47%, 38–59% | 38%, 30–46% | 54%, 45–67% |
| DLCO $\leq$ 35% | 16.5% (40/243) | 20.5% (15/73) | 30.4% (21/69) | 3.96% (4/101) |
| 6MWD, m (median, IQR) | 330, 240–424 | 328, 210–445 | 260, 200–340 | 384, 314–456 |
| Nadir % SpO2 (median, IQR) | 87%, 82–91% | 84%, 80–88% | 85%, 82–89% | 91%, 85–95% |
| GAP Stage I | 36.8% (102/277) | 36.8% (32/87) | 23.8% (20/84) | 47.2% (50/106) |
| GAP Stage II | 43.3% (120/277) | 41.4% (36/87) | 45.2% (38/84) | 43.4% (46/106) |
| GAP Stage III | 19.9% (55/277) | 21.8% (19/87) | 30.9% (26/84) | 9.43% (10/106) |
| Antifibrotics (%) | 39.5% (110/298) | 24.2% (23/95) | 31.0% (26/84) | 58.0% (69/119) |
| Anti-acid medications (PPI) (%) | 58.8% (174/296) | 66.3% (63/95) | 47.6% (40/84) | 60.7% (71/117) |
| NAC (%) | 9.06% (27/296) | 3.16% (3/95) | 4.76% (4/84) | 17.1% (20/117) |
| NIV (CPAP/Bipap) | 4.62% (12/260) | 19.3% (11/57) | 1.19% (1/84) | 0 |
| Transplant | 1 tx, 9 listed | 0 tx, 6 listed | 1 tx, 1 listed | 0 tx, 2 listed |

CCI: Charlson Comorbidity Index; MRC: medical research council; PPI; proton pump inhibitor; NAC: N-acetyl cysteine; NIV: non-invasive ventilation; CPAP: continuous positive airway pressure; and BiPAP; bi-level positive airway pressure.

### 3.2. Palliative Care Delivery—Increased Symptom Management and ACP

Overall, 63% (188) received PC as per study definition: 98% (93) in EC, 94% (79) in BC, and 13% (16) in QC ($p < 0.001$; Table 2). Twenty percent were referred to specialist PC (3% EC, 20% BC, and 21% QC). Additionally, in BC cohort, 54% (45) were discussed

in Palliative-ILD-Psychology (PIP) multidisciplinary team meetings where specialist PC advice was received. In QC, 21% of patients were referred to the specialist PC clinic based on criteria described but only 13% received PC based on study criteria. Significant differences were noted across cohorts. High rates of non-pharmacologic dyspnea strategies (pacing or energy conversation) were used in EC (98%) and BC (94%) in contrast to 2% recorded in QC ($p < 0.001$). In the QC cohort, non-pharmacologic management of dyspnea (and other symptoms) was routinely addressed at each PC visit but not documented. Similarly, higher rates of oxygen use noted in EC (71%) and BC (87%) compared to 24% in QC ($p < 0.001$). Forty-five percent of EC and BC patients were on opiates for dyspnea and 23% of QC ($p < 0.001$) (Table 2, Figure 1A). Data from patients seen in the QC specialist PC clinic are analyzed separately in Supplement Table S4.

**Table 2.** Outcomes (total population alive and deceased).

| V1 and Any Visit | All Cohorts (*n* = 298) | | EC (*n* = 95) | | BC (*n* = 84) | | QC (*n* = 119) | | *p* Value |
|---|---|---|---|---|---|---|---|---|---|
| | V1 | Any visit | V1 | Any visit | V1 | Any visit | V1 | Any visit | |
| Palliative care (%) | 54% (161/298) | 63% (188/298) | 91.6% (87/95) | 98% (93/95) | 86.9% (73/84) | 94% (79/84) | 0.84% (1/119) * | 13.4% (16/119) * | <0.001/<0.001 |
| **I. Symptom Assessment/Documentation (%Y, tool)** | | | | | | | | | |
| MRC | 97% (289/298) | 97.7% (291/298) | 95.8% (91/95) | 96.8% (92/95) | 96.4% (81/84) | 97.6% (82/84) | 97.5% (116/119) | 99.2% (118/119) | 0.8/0.4 |
| Other | 64.8% (193/298) | 93.9% (201/214) | 91.6% (87/95) MDDS | 93.7% (89/95) MDDS | ILD tool # NR | | UCSD: 89.1% (106/119) | UCSD: 94.1% (112/119) | 0.6/ 0.05 |
| **II. Symptom Management (%Y)** | | | | | | | | | |
| Pacing or energy conservation | 53.7% (160/298) | 58.4% (174/298) | 91.6% (87/95) | 97.9% (93/95) | 86.9% (73/84) | 94.0% (79/84) | 0% (0/119) * | 1.68% (2/119) * | <0.001/<0.001 |
| Oxygen | 33.2% (99/298) | 55.7% (166/298) | 48.4% (46/95) | 70.5% (67/95) | 52.4% (44/84) | 86.9% (70/84) | 7.56% (9/119) | 24.4% (29/119) | <0.001/<0.001 |
| Opiates (for dyspnea) | 7.71% (23/298) | 28.2% (84/298) | 14.7% (14/95) | 45.2% (43/95) | 10.7% (9/84) | 45.2% (38/84) | 0.840% (1/119) | 22.7% (27/119) | <0.001/<0.001 |
| **III. ACP (A–D)** | | | | | | | | | |
| A. Patient understanding Goals/values/ caregiver engagement (any 1) | 58.4% (174/298) | 66.1% (197/298) | 93.7% (89/95) | 97.9% (93/95) | 100% (84/84) | 100% (84/84) | 5% (1/119) | 13.4% (16/119) | <0.001/<0.001 |
| B. Patient preferences documented | 31.2% (93/298) | 52.3% (156/298) | 67% (64/95) | 83.1% (79/95) | 33.3% (28/84) | 71.4% (60/84) | 5.88% (1/119) | 14.2% (17/119) | <0.001/<0.001 |
| DNI/DNR/non-ICU (medical care) only | 19.1% (57/298) | 43.6% (130/298) | 48.4% (46/95) | 62.1% (59/95) | 11.9% (10/84) | 67.8% (57/84) | 0.84% (1/119) | 11.8% (14/119) | <0.001/<0.001 |
| C. GOC form | 41.6% (124/298) | 49.3% (147/298) | 43.2% (41/95) | 58.9% (56/95) | 97.6% (82/84) | 97.6% (82/84) | 0.84% (1/119) * | 7.56% (9/119) * | <0.001/<0.001 |
| D. EOL discussion | 29.9% (89/298) | 47.7% (142/298) | 71.6% (68/95) | 84.2% (80/95) | 22.6% (19/84) | 63.1% (53/84) | 1.68% (2/119) * | 7.56% (9/119) * | <0.001/<0.001 |
| **IV. Palliative assessments (%Y)** | | | | | | | | | |
| PC Specialist consultation | 1.68% (5/298) | 19.5% (58/298) | 0% (0/95) | 3.16% (3/95) | 2.38% (2/84) | 20.2% (17/84) | 2.52% (3/119) | 21% (25/119) | 0.7 |
| Palliative multidisciplinary discussion | N/A | N/A | N/A | N/A | 11.9% (10/84) | 54% (45/84) | N/A | N/A | N/A |

* QC—some ACP and PC data missing (patients may have been seen in community PC).

Significant differences were also noted in ACP practices across cohorts. Communication regarding disease, prognosis and planning was initiated in 98% EC and 100% BC patients compared to 5% in QC at first ILD visit ($p < 0.001$) (Table 2). All EC and BC patients had at least one component of ACP discussions by last visit compared to 14% of QC, all of whom were all seen in the specialist PC clinic (Figure 1C). ACP documentation was not undertaken in 87% QC patients. First-visit EOL discussions occurred in 30% of the overall cohort, among 72% EC, 23% BC, and 2% QC patients (Table 2). For QC, EOL care discussions typically occurred late when patients started receiving PC at home.

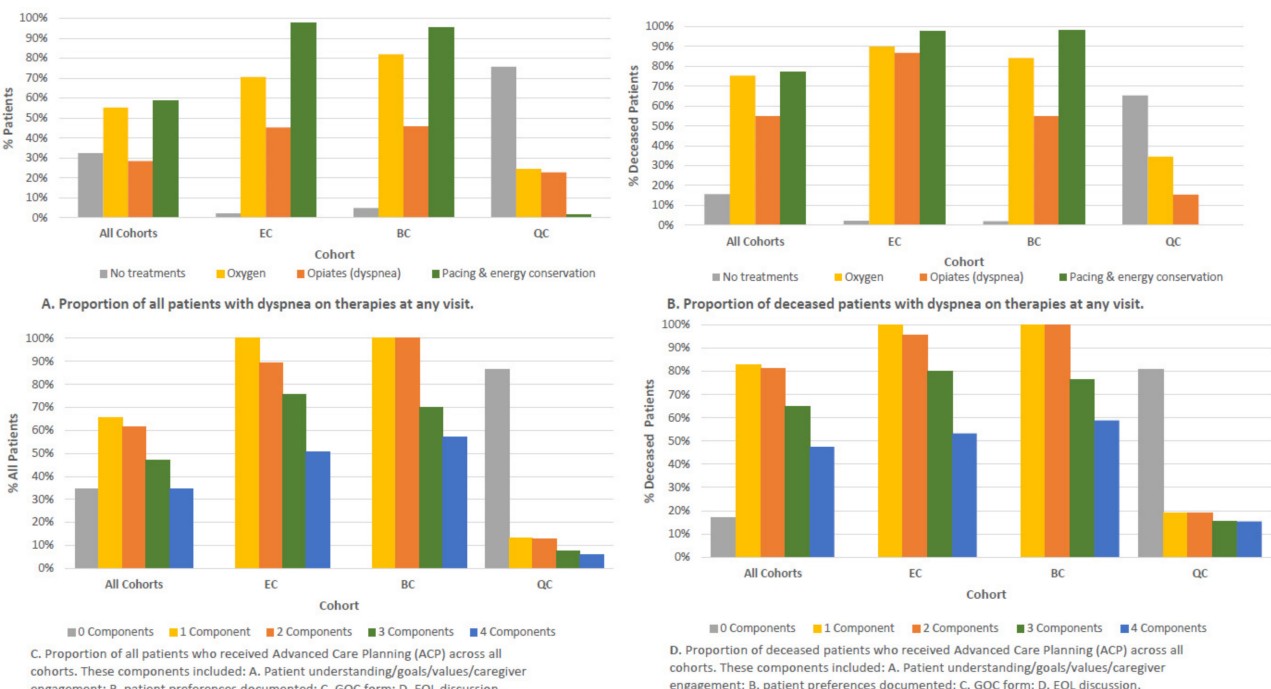

**Figure 1.** (**A**–**D**) panel plots: (**A**): dyspnea management in all patients; (**B**): dyspnea management in decedents; (**C**): ACP documentation in all patients; and (**D**): ACP documentation in decedents.

More patients in EC and BC models received supportive services compared to QC, including allied health referrals (62% EC, 37% BC and 5% QC) and home care services (69% EC, 67% BC, and 0% QC) (Supplement Table S1). Overall, 41% were referred to pulmonary rehabilitation (EC 68%, BC 86%, and QC 14%) and 43% to ILD support groups (45% EC, 96% BC, and 3% QC; Supplement Table S2). Caregiver engagement occurred in 64% (96% EC, 99% BC, and 13% in QC) (Supplement Table S2).

### 3.3. Improved EOL Care

One hundred and twenty two patients died during the study giving a mortality rate of 41% (122; consisting of 45 EC (47%), 51 BC (61%), and 26 QC (22%)) (*p* < 0.001). Median (IQR) follow-up time was 16 (5–28) months (Table 3; Supplement Table S3). Overall, 76% of decedents had received PC at some point during their illness trajectory: 91% (41) in EC and 92% (47) in BC. QC EOL data is available only for 27% (7/26) decedents who were seen in the specialist PC clinic. There were significant differences across models (Table 3 and Figure 1B). At EOL, overall 66% (81) were on oxygen and 35% (43) were on opioids for dyspnea, initiated 15 and 5 (median) months before death, respectively. Significantly more decedents in EC and BC were on supplemental oxygen and opioids than QC (Oxygen: *p* = 0.04; Opioids: *p* < 0.001) (Table 3). Ninety-one percent EC and 92% BC decedents were taught non-pharmacologic dyspnea strategies (pacing, energy conservation); there were no data recorded in QC (*p* < 0.001). Supplement Figure S1 displays a timeline of the PC therapies decedents received across all cohorts. Amongst the QC patients who were referred to specialist PC clinic (21%), only seven died, and most of them received symptom management (Supplement Table S4).

**Table 3.** Clinic-based end of life palliative interventions (decedents)—last 12 months before death.

| | All Cohorts | EC | BC | QC | *p*-Value |
|---|---|---|---|---|---|
| Deceased of total | 40.9% (122/298) | 47.4% (45/95) | 60.7% (51/84) | 21.8% (26/119) | <0.001 |
| Palliative care (%) | 76.2% (93/122) | 91.1% (41/45) | 92.1% (47/51) | 19.2% (5/26) | <0.001 |
| Interval between last visit and death (median months, IQR) | 3, IQR 1–6 | 2, IQR 1–5 | 3, IQR 1–5 | 3.5 *, IQR 1.25–15.5 | 0.01 |
| Median #EOL visits (last 6 months), IQR | 1, 0–2 | 1, 1–2 | 1, 1–2 | 0 *, 0–2 | 0.3 |
| Symptom Management | | | | | |
| Pacing or Energy conservation | 72.1% (88/122) | 91.1% (41/45) | 92.2% (47/51) | 0% (0/26) * | <0.001 |
| Oxygen (%) | 66.4% (81/122) | 82.2% (37/45) | 76.5% (39/51) | 19.2% (5/26) * | 0.04 |
| *Initiated (median months, IQR)* | 15, IQR 6–27 | 21, IQR 9–31 | 10, IQR 5–21 | NR | <0.001 |
| Opiates for dyspnea (%) | 35.2% (43/122) | 51.1% (23/45) | 39.2% (20/51) | 15.4% (4/26) * | 0.01 |
| *Initiated (median months, IQR)* | 5, IQR 2–9 | 5, IQR 2–12 | 4, IQR 2–8 | N/A | 0.1 |
| ACP (A-D) | | | | | |
| A. Documentation of patient understanding/goals/ values/CG (any) | 80.3% (98/122) | 93.3% (42/45) | 100% (51/51) | 19.2% (5/26) * | <0.001 |
| B. Patient preferences documented | 68.8% (84/122) | 93.3% (42/45) | 72.5% (37/51) | 19.2% (5/26) * | 0.001 |
| Preferred type of Care (DNR/DNI/Others) | 63.9% (78/122) | 80% (36/45) | 72.5% (37/51) | 19.2% (5/26) * | <0.001 |
| Preferred location of death | Home 50% (61/122) Hospice 0.001% (4/122) Hospital 5.7% (7/122) | Home 60% (27/45) Hospice 8.8% (4/45) Hospital 13.3% (6/45) | Home 66.6% (34/51) Hospice 0 Hospital 2% (1/51) | NR | |
| C. GOC form | 61.4% (75/122) | 60% (27/45) | 86% (44/51) | 15.4% (4/26) * | <0.001<0.001 |
| D. EOL disc | 54.1% (66/122) | 73.3% (33/45) | 62.7% (32/51) | 3.8% (1/26) * | 0.001 |
| Palliative assessments (%Y) | | | | | |
| PC Specialist Consultation | 13.9% (17/122) | 6.6% (3/45) | 17.6% (9/51) | 19.2% (5/26) * | 0.2 |
| Palliative Multidisciplinary Discussion | N/A | N/A | 49.0% (25/51) | N/A | |

* QC data is an underestimate; data located in paper records was not included. Some patients would have received PC in the community but that data is not available. GOC = goals of care.

Overall, 80% had ACP discussions, with 69% patient preferences recorded and 64% with documented do not resuscitate/intubate (DNR/DNI) orders; 59% had preferred location of death recorded, and 54% had end of life discussions (Table 3 and Figure 1D). While 93% and 100% of EC and BC decedents, respectively, had ACP discussions, 81% of QC did not (Figure 1D). Home or hospice was the preferred location of death in 53% (65) of patients, when documented (Table 3). Overall, 59% died at home or hospice and 38% in hospitals (1 in intensive care) (Table 4). Fifty-nine% in EC and 58% in BC models died at their preferred location of home or hospice. In the two sites that recorded preference for location of death (EC and BC), there was "fair" agreement between the patient preference and location of death in the EC cohort (k = 0.29) and no agreement at the BC site (k = −0.11). Respiratory cause of death was reported in 95% EC, 80% BC, and 46% of QC decedents (not reported in 50% of QC).

**Table 4.** Location of Death and concordance (actual/preferred location).

| Location of Death | All Cohorts | EC | BC | *p*-Value |
|---|---|---|---|---|
| Home Hospice | 47.9% (46/96) 11.5% (11/96) | 42.2% (19/45) 20% (9/45) | 52.9% (27/51) 3.92% (2/51) | 0.05 |
| Home & Hospice concordance rate (actual: preferred) | 57:98 | 28:48 | 29:50 | |
| Hospital ICU | 37.5% (36/96) 1.04% (1/96) | 37.8% (17/45) 2.22% (1/45) | 37.3% (19/51) 0% (0/51) | 0.82 |
| Hospital concordance rate (actual/preferred) | 36:10 | 17:7 | 19:3 | |
| Location of Death Concordance Kappa (95% CI) | 0.002 (−0.146–0.170) | 0.29 (−0.02–0.51) | −0.11 (−0.20–0.15) | |

## 4. Discussion

Improving access to PC and EOL care is an important goal in management of IPF patients who experience serious health-related suffering throughout their illness and especially near the end of life. Comprehensive patient centered care in IPF and ILD, therefore, requires a focus on symptoms, support, and ACP in practice [14]. Such a comprehensive approach extends routine care beyond making accurate and timely diagnosis and instituting disease-specific medical management to provide comprehensive support for living and coping well [15]. Lee and colleagues' systematic review identified diverse ILD/IPF patient needs ranging from psychological, physical, daily living, information, and education to preparing for EOL [2]. Patients also desire organized follow-up and better care coordination [16]. Integration of a PC approach in routine care by ILD experts, general pulmonologists, internists, and family physicians offers the possibility to meet these needs while simultaneously delivering evidence-based disease-targeted therapies. There is universal agreement on the need for PC access, but delivery of PC remains challenging. Currently, the gold-standard model of PC delivery is unknown [10,17]. Quill and colleagues suggest that the PC needs for many chronic diseases can be safely met through a primary PC model where the healthcare professionals themselves provide a PC approach with specialist consultations reserved only for challenging issues. There are no assessments of this approach in ILD/IPF. Our study addresses this evidence gap by describing primary PC and specialist PC referral models in real-world IPF cohorts and the impact of each model on care.

Our work suggests that a primary PC approach is feasible in IPF when adequate resources and infrastructure are in place and can result in early and concurrent delivery of PC. Whereas reliance on specialist PC referral may result in fewer patients receiving PC and a significant delay in PC interventions. EC model sought specialist PC for hospice referrals and the collaborative PIP multidisciplinary team in BC integrated psychology and specialist PC with pulmonologists and community teams to meet PC needs of IPF/ILD patients [13]. Specialist PC referral rates observed in QC are similar to those described in the literature, where 15% were referred to PC [18]. When a primary PC approach is undertaken by pulmonologists, more patients receive symptom management and ACP early in the disease course, concurrent with IPF-specific care (antifibrotics and transplant assessments). It is particularly striking to note the high uptake of non-pharmacologic dyspnea management strategies seldom used in traditional clinical practice. Additionally, primary PC models also improved access to rehabilitation, support groups, and home care services (where available) and led to early engagement of family caregivers. EOL care in IPF is generally poor with inadequate and delayed symptom management; there is a lack of timely EOL conversations, less family engagement, and limited support, with majority of patients dying in less desired acute care settings [18–20]. Primary PC models with systematic ACP improve EOL care, with > 90% of EC and BC decedents receiving PC. This is a substantial improvement from current practice, where PC is not integrated. Systematic ACP led to more patients dying in their preferred location in the primary PC models. Lack of EOL data in the specialist PC model (QC) hampered further comparisons. Even though the specialist PC referral model (QC) had fewer patients receiving PC, the majority of those referred received symptom management and engaged in ACP.

From a patient perspective, the primary PC model has many advantages. Many patients with progressive dyspnea, fatigue, and high flow oxygen may not be willing or able to attend another clinic appointment. This was noted in a feasibility trial of PC referral in IPF patients, where 60% were reluctant to attend PC clinics outside their regular ILD visits. In addition, patients' misperceptions and lack of knowledge about the role of PC may result in many declining such referrals [21]. Primary PC approach is particularly important for IPF patients, where unpredictable deterioration and death may occur early in the disease without the opportunity for timely referral and subsequent assessment by PC specialists [22]. Currently, there are no standardized specialist PC referral criteria in IPF, and reliance on physiological indicators of severity may not reflect patients' symptoms and

needs. Moreover, symptoms and needs change independent of physiology over time. In QC, PC referral criteria included dyspnea, oxygen use, and disease progression, generally based on physiology that may not detect many unrelated needs. PC co-management with referral to co-located PC clinics offers another option but may not reach all patients who need PC. ILD experts in this study perceived some barriers to this approach, including time required to coordinate, communicate, and discuss PC referral with their patients [23].

Our study also shows that primary PC models can improve dyspnea assessment and management, a research priority in IPF [24]. A meta-analysis concluded that breathlessness services that offer a multidisciplinary, holistic, and patient-centered approach, including non-pharmacologic therapies, allied health assessments, and education, were effective in reducing symptom distress [25]. Outside of a few such breathlessness services, and clinical studies of short duration, there are limited real-world examples of integrated dyspnea management in ILD [26]. Non-pharmacologic dyspnea therapies are seldom prescribed in IPF [27], even though they are core aspects of many integrated breathlessness services [20,26,28]. In the absence of such a comprehensive approach in most ILD clinics, dyspnea is not well managed [20,27]. Adoption of a primary PC approach by ILD physicians facilitates early identification and management of dyspnea, including high rates of use of non-pharmacologic dyspnea strategies (pacing, energy conservation), as reported in EC and BC. The use of fans was also reported in >90% of BC cohort (data not shown). Higher proportion of patients in our study were on ambulatory oxygen overall (56%) when compared to around 20% in clinical trials cohorts and other registry studies [29,30]. This is particularly important given recent evidence that suggests supplemental oxygen can improve quality of life and symptoms in fibrotic ILD in the short term [31]. Similarly, early use of opioids is noted in both the primary PC models in contrast to delayed use described in the literature; opioids are mostly initiated in the last week of life in IPF in absence of early PC [19]. The majority of the decedents in the primary PC models received PC in the last year of life, with high rates of symptom therapies, including non-pharmacologic strategies and oxygen use. Opioids were started at median 15 months before death. Of note, 80% of QC decedents had not been reviewed by the PC specialist and 50% of BC decedents died before their case was reviewed in PIP-MDT rounds. This highlights the unpredictable disease course with rapid progression and underscores the need for an early integrated palliative care approach. Use of a primary PC approach enabled PC therapies to be instituted in BC decedents without specialist PC input.

Primary PC models also support early and ongoing ACP, an important patient perceived gap in IPF communication. ACP conversations seldom occur in IPF practice, even though most patients want these discussions early in the course [17,19,27,32]. Studies suggest that early and integrated ACP can improve care and communication, help patients and families prepare for EOL, relieve anxiety, and facilitate patient preferences for place of death [8,33,34]. Literature shows that EOL care is poor and care decisions are delayed and infrequent. In a Finnish cohort, EOL decisions were recorded in only 32% (DNR orders in 57%), and 42% were within the last 3 days of life [19]. Our data represent a significant improvement, with patient preferences and DNR orders documented in >70% of decedents by their last visit in EC and BC (3 months before death). Decedents in our study had early ACP with EOL discussions. Literature suggests that most IPF patients and their families prefer to have early ACP conversations [2,35]. Majority of our IPF patients engaged in early ACP, and the timing of these EOL discussions in EC and BC models was based on physician assessment of patients' prognosis. This was based on patient performance status and activities of daily living tolerance; respiratory symptoms' severity; oxygen needs; and nadir desaturation with activity, pulmonary function, exercise tolerance, and their serial change. In addition, greater concordance between patients' preferred and actual location of death in the primary PC models suggests that this model can facilitate patient-centered EOL goals, although we appreciate that patient preferences cannot always be fulfilled. Our work shows that ACP discussions can be started early as per patients' wishes and continue over time without waiting for specialist PC referral, a precious resource that may be better

reserved for complex cases. Future studies can examine the role of each ACP component and their relative impact on outcomes. recent study noted that pulmonologists perceived less confidence in ACP for IPF patients when compared to lung cancer [36]. In order to integrate ACP into IPF care, clinicians must develop ACP competency and willingness to adopt and integrate PC approach in care. Previous research in cancer patients has shown that gender differences in attitudes and behavior can impact PC referral and EOL care delivery [37]. For example, men are more likely to prefer life-sustaining therapies than women who are also more likely to prefer PC approach. This may also play a role in decision-making and care of IPF patients and needs to be explored in future studies.

*Limitations*

The study populations in the three models compared were heterogeneous. The cohort was male predominant, as seen in all IPF cohorts, and thus may have an overrepresentation of patients with more impaired baseline lung function [38]. QC had less severe patients; this may have contributed to reluctance for ACP discussions and PC referral. All ILD centers determined their own protocols and tools of PC needs assessment and management. The care teams' composition was also variable. QC data was incomplete due to lack of access to some paper charts as multiple services were engaged in providing PC, especially at EOL (community PC specialist, family doctors, or hospital-based PC services). In QC, an important factor to consider is that some patients refused the offered treatments (e.g., opioids) and were counted as receiving no treatment but essentially represent patients' choices. Despite heterogeneity in models and approach, there is congruence in some aspects of PC provided, including time to initiation of various strategies between both the models primary palliative care approach (EC and BC). Our study did not assess the impact of PC therapies on symptoms, HRQoL, patient and caregiver satisfaction, or healthcare utilization rates.

## 5. Conclusions

Access to PC and good EOL care are unmet needs in IPF. The primary PC model is feasible in IPF and results in improved access to palliative services. Integration of PC into ILD clinics can lead to early implementation of symptom management, ACP discussions, and better EOL care as per patient preferences. Based on our data, specialist PC referral while feasible may result in fewer patients receiving PC overall. Major limitations to primary PC models include lack of PC competency in care providers and needed infrastructure support in clinics and the communities. Randomized controlled clinical trials are needed to further define the ideal PC model in IPF/ILD.

**Supplementary Materials:** The following are available online at https://www.mdpi.com/article/10.3390/app11199028/s1, Table S1: Model description; Table S2: Antifibrotic therapies, Symptom Management and Advanced Care Planning; Table S3: Duration of Follow-up; Table S4: Queen's PC interventions; Figure S1: Key event timelines from death. This includes median months of time from death for decedents only. Dates for opiates and oxygen initiation were not recorded at QC.

**Author Contributions:** Data curation S.Y., N.R., H.A.; Formal analysis S.Y., J.A.B., M.K.; Project administration S.Y.; Methodology I.H., O.M.-M., S.L.B., H.A., M.K.; Writing-original draft M.K.; Writing-review & editing S.Y., J.A.B., J.R.-E., M.M., N.R., G.A., S.A., E.B., I.H., O.M.-M., S.L.B., H.A., M.K.; Investigation G.A., S.A., M.M.; Conceptualization M.K.; Resources M.K.; Supervision M.K. All authors contributed to drafting, revising, and finalizing the manuscript. All authors have read and agreed to the published version of the manuscript.

**Funding:** This research was funded by Alberta Respiratory Center, grant number [SC ZJ169]. And The APC was funded by ILD Clinic donations.

**Institutional Review Board Statement:** The study was conducted according to the guidelines of the Declaration of Helsinki, and approved by The Health Research Ethics Boards in Alberta (REB Pro00082981), Ontario (HSREB 6025437), and Bristol, UK (IRAS 254044).

**Informed Consent Statement:** The study included deceased IPF patients, therefore the consent requirement was waived.

**Data Availability Statement:** Provided at Supplementary Materials.

**Acknowledgments:** Study data were collected and managed using REDCap* electronic data capture tools hosted by the Women & Children's Health Research Institute at the University of Alberta. *Paul A. Harris, Robert Taylor, Robert Thielke, Jonathon Payne, Nathaniel Gonzalez, Jose G. Conde, Research electronic data capture (REDCap)—A metadata-driven methodology and work-flow process for providing translational research informatics support, J Biomed Inform. 2009 Apr;42(2):377-81.

**Conflicts of Interest:** The authors declare no conflict of interest.

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
