# Peer review of "Comparison of Palliative Care Models in Idiopathic Pulmonary Fibrosis"

_applsci, doi:10.3390/app11199028_

Round 1

Reviewer 1 Report

The manuscript written by Sarah Younus etc demonstrates the critical importance of palliative care to improve vital signs and survival of IPF patients, comprising different models. The topic is very interesting and could be useful for the medical and scientific communities. There are some comments which could help to improve the manuscript for the publication in Applied Sciences.

  1. The authors used only male subjects for the study. Despite the obvious reason, it should be justified in discussion why you did not use females.  The literature data about sex-related differences in pulmonary fibrosis should be added. Also, adding a short discussion about palliative care for women would be useful.
  2. Figure 2: error bars should be added to the graphs. 
  3. There are some grammatical errors in the text. 
  4. Please check all the references. Some of them do not correspond to numbering, not designed correctly and without accordance with the journal requirements.

Author Response

Response: We thank the reviewer for the kind comments and for highlighting the critical importance of the submitted work. We also believe that a comparison of models will help the medical and scientific community move forward in a direction to positively impact patient quality of life in IPF.

Comment 1: The authors used only male subjects for the study. Despite the obvious reason, it should be justified in discussion why you did not use females.  The literature data about sex-related differences in pulmonary fibrosis should be added. Also, adding a short discussion about palliative care for women would be useful.

Response 1: We did not purposefully exclude female patients. IPF is more common in males (2:1), in our cohort 74% were males which is not very different from other published studies (BMC Pulm Med. 2019 Nov 27;19(1):222. doi: 10.1186/s12890-019-0994-4). We have added a few sentences to reflect this and the issue of gender difference in palliative care. Thank you for the valuable comment.

 Comment 2: Figure 2: error bars should be added to the graphs.

 Response 2:  There is only Figure 1; we did not submit a Figure 2 in the main manuscript. We do not believe that the error bars are needed here, it is a report on the data without inference. It will make the graphs too busy and make it difficult to read. But we are happy to add them if the editors feel strongly about it.

 Comment 3: There are some grammatical errors in the text.

Response 3: we have checked spelling and grammar and made corrections where needed.

Comment 4: Please check all the references. Some of them do not correspond to numbering, not designed correctly and without accordance with the journal requirements.

Response 4: The references have been reviewed.

Reviewer 2 Report

Comments to Author

The authors suggested that primary PC approach for IPF is feasible in ILD clinics with concurrent disease management and can improve access to symptom management, ACP PC and EOL care. Although this paper was well written, sufficient improvements for following issues are needed for the acceptance to the Applied Sciences.

Major comments

Your study attempts to clarify the important issue that palliative care for IPF patients is often tends to be insufficient and/or late, compared to that of patients with lung cancer.

In your study, I would like you to sufficiently explain the appropriate timing of decisions of EOL of IPF patients. I understand that the timing of decisions of EOL of IPF patients is more difficult than that of patients with lung cancer. Subjects in your study includes many IPF cases of GAP stage 2 or lower. Due to the heterogeneous clinical behavior of IPF patients, too early decision of EOL may have risk to discourage physicians and patients from treatment for patients with reversible IPF. We understand that the appropriate timing of EOL decisions depends on performance status, ADL, respiratory symptoms, pulmonary function, and exercise tolerance and their serial change. Please clarify that issue in your study.

Author Response

Comments to the Author

The authors suggested that primary PC approach for IPF is feasible in ILD clinics with concurrent disease management and can improve access to symptom management, ACP PC and EOL care. Although this paper was well written, sufficient improvements for following issues are needed for the acceptance to the Applied Sciences.

Response:  Thank you for the kind comments.

Major comments: Your study attempts to clarify the important issue that palliative care for IPF patients is often tends to be insufficient and/or late, compared to that of patients with lung cancer.

In your study, I would like you to sufficiently explain the appropriate timing of decisions of EOL of IPF patients. I understand that the timing of decisions of EOL of IPF patients is more difficult than that of patients with lung cancer. Subjects in your study includes many IPF cases of GAP stage 2 or lower. Due to the heterogeneous clinical behavior of IPF patients, too early decision of EOL may have risk to discourage physicians and patients from treatment for patients with reversible IPF. We understand that the appropriate timing of EOL decisions depends on performance status, ADL, respiratory symptoms, pulmonary function, and exercise tolerance and their serial change. Please clarify that issue in your study.

Response: Thank you for your comment and raising the issue. You are correct, IPF disease trajectory is unpredictable and at presentation it is difficult to identify patients who will have a rapid decline. For example,  Liang et al showed in their study that up to 34% may end up requiring ICU care for respiratory failure within four months of diagnosis/first clinic visit and nearly 50% had an ICU admission before the third clinic visit (Liang Z. Journal Of Palliative Medicine 2017; 20(2). DOI: 10.1089/jpm.2016.0258). Given this high number we tend to initiate ACP early. As part of ACP, we discuss EOL preferences as well.

IPF is NOT a reversible disease even at diagnosis (even if made early). It is a prototype of progressive-fibrosing ILD. None of the current treatments can reverse or stop the progression, they only slow it down. Thus, antifibrotic therapy shares similarities with palliative chemotherapy in cancer. They do not treat symptoms or improve quality of life. Palliative care is therefore, presented as an integrated approach rather than a dichotomous approach (instituting PC after curative therapies). The American Thoracic Society IPF 2011 guidelines also suggest early PC and early ACP along with disease-modifying therapies.

We do not believe that discussing EOL early dissuades patients or physicians from pursuing disease-modifying therapies. This is demonstrated in our study, 40% were on antifibrotics (disease-specific therapies) at baseline and with the integrated PC approach in two models, it increased to 63% in the overall cohort. This suggest that use of PC or early EOL discussions does not lead to reduction of disease-specific therapies. In addition, these discussions are always individualized and incorporate patient wishes and preferences for the same.

We have added a few sentences in the discussion to indicate our timing of EOL conversations.

Thank you once again for taking the time to review our work and provide suggestions to strengthen its value.

Round 2

Reviewer 1 Report

The manuscript was significantly improved and could be accepted in its present form.

Reviewer 2 Report

Author have responded to my question correcty.